# Sonication-Assisted Method for Decellularization of Human Umbilical Artery for Small-Caliber Vascular Tissue Engineering

**DOI:** 10.3390/polym13111699

**Published:** 2021-05-22

**Authors:** Chih-Hsun Lin, Kai Hsia, Chih-Kuan Su, Chien-Chin Chen, Chang-Ching Yeh, Hsu Ma, Jen-Her Lu

**Affiliations:** 1Division of Plastic and Reconstructive Surgery, Department of Surgery, Taipei Veterans General Hospital, Taipei 11217, Taiwan; chlin12@vghtpe.gov.tw (C.-H.L.); hkay1008@gmail.com (K.H.); quroboros376@gmail.com (C.-K.S.); sma@vghtpe.gov.tw (H.M.); 2Department of Surgery, School of Medicine, National Yang Ming Chiao Tung University, Taipei 11221, Taiwan; 3Department of Pathology, Ditmanson Medical Foundation Chia-Yi Christian Hospital, Chiayi 600, Taiwan; hlmarkc@gmail.com; 4Department of Cosmetic Science, Chia-Nan University of Pharmacy and Science, Tainan City 71710, Taiwan; 5Department of Obstetrics and Gynecology, Taipei Veterans General Hospital, Taipei 11217, Taiwan; ccyeh39@gmail.com; 6Department of Obstetrics and Gynecology, National Yang Ming Chiao Tung University, Taipei 11221, Taiwan; 7Department of Nurse-Midwifery and Women Health, National Taipei University of Nursing and Health Sciences, Taipei 11219, Taiwan; 8Institute of Clinical Medicine, National Yang Ming Chiao Tung University, Taipei 11221, Taiwan; 9Department of Surgery, Medicine & Pediatrics, School of Medicine, National Defense Medical Center, Taipei 11490, Taiwan; 10Section of Pediatric Cardiology, Department of Pediatrics, Taipei Medical University Hospital, Taipei 11031, Taiwan

**Keywords:** sonication decellularized human umbilical artery, ultrasonic decellularization, extracellular matrix, biocompatibility, recellularization

## Abstract

Decellularized vascular grafts are useful for the construction of biological small-diameter tissue-engineered vascular grafts (≤6 mm). Traditional chemical decellularization requires a long treatment time, which may damage the structure and alter the mechanical properties. Decellularization using sonication is expected to solve this problem. The aim of this study was to develop an effective decellularization method using ultrasound followed by washing. Different power values of sonication at 40 kHz were tested for 2, 4, and 8 h followed by a washing procedure. The efficacy of sonication of decellularized human umbilical artery (sDHUA) was evaluated via DNA content, histological staining, mechanical properties, and biocompatibility. The sDHUAs were further implanted into rats for up to 90 days and magnetic resonance angiography (MRA) was performed for the implanted grafts. The results demonstrated that treatment of human umbilical artery (HUA) by sonication at ultrasonic power of 204 W for 4 h followed by washing for 24 h in 2% SDS buffer could eliminate more than 90% of cells and retain similar mechanical properties of the HUA. Recellularization was assessed by scanning electron microscopy (SEM), which indicated that sDHUA provided niches for human umbilical vein endothelial cells (HUVECs) to reside, indicating in vitro cytocompatibility. Further implantation tests also indicated the fitness of the sonication-treated HUA as a scaffold for small-caliber tissue engineering vascular grafts.

## 1. Introduction

Cardiovascular diseases (CVDs) remained as the leading cause of human death globally. These patients suffered from vascular occlusive disease at heart, brain, peripheral limb, etc. The obstruction of vascular blood flow to these vital organs could result in sudden death, stroke, or limb ischemia [1]. In severe cases refractory to medical treatment, angioplasty, or stent placement, bypass surgery with vascular graft is the final option to reconstitute blood flow [2]. Autologous vessels are the preferred bypass materials but about one-third of patients do not have suitable donor vessel [3]. Prosthetic vascular grafts, such as polyethylene terephthalate (Dacron) and expanded polytetrafluoroethylene (ePTFE), could not provide long-term patency when the diameter is under 6 mm [4,5]. Thus, development of innovative technologies focusing on small-caliber vascular grafts is indeed required for vascular reconstructive surgery [6]. Vascular tissue engineering represents one of the emerging research fields aimed to fabricate small-caliber vascular grafts utilizing tissue engineering technique [7].

Development of small-caliber (<6 mm) arterial scaffolds for vascular tissue engineering research can be approached in different ways. Of these, using decellularized extracellular matrix (ECM) scaffolds, such as tissues from humans or other species, has gained broad attention. The decellularization process usually involves physical, chemical, or enzymatic methods to remove cellular components from native vascular tissues while leaving ECM in the tubular structures [8]. Efficient decellularization is a major requirement in the process because inadequate decellularization could markedly influence pro-inflammation and remodeling [9]. Thoroughly decellularized scaffolds reduce immunogenicity [10]. In addition, the tubular biological scaffolds preserve the fibril elements and biomolecules inside. The scaffolds theoretically maintain biochemical and biomechanical properties, which are beneficial in cellular recruitment and can withstand cyclic pressure from blood flow. However, the detergents or enzymes used in decellularization may also damage the components of the extracellular matrices and lead to alterations in microstructure, thereby affecting the biological and biomechanical performance of the scaffolds. Thus, identifying an ideal decellularization process is the first step in the development of ECM scaffolds.

To overcome the drawbacks of current decellularization protocols, researchers continue to develop new methods to decrease treatment duration, reduce exposure to chemical or organic substances, and reduce tissue damage [11]. For example, supercritical fluid (carbon dioxide) without detergents has been used for the decellularization of aortic tissues [12]. High hydrostatic pressure without chemical agents has been used for corneas and blood vessels [13,14]. Recently, Azhim et al. developed a sonication protocol with sodium dodecyl sulphate (SDS) for the decellularization of aortic tissues. This indicated that sonication treatment had the ability to complete decellularization of the extracellular matrix [15,16]. In addition to vascular tissue, Say et al. showed effective perfusion decellularization of the kidney by SDS and concomitant sonication. Removal of cellular content while preserving various renal structures (glomerulus, tubular structure, and blood vessels) was noted [17]. In addition, sonication has an effect on the decellularization of cartilaginous tissue while maintaining its biomechanical strength [18].

The effect of sonication in decellularization depends primarily on the cavitation phenomenon, in which bubbles form to physically dissociate molecules [19]. As more cavitation bubbles are formed during sonication, many molecules are dissociated [13]. Several factors, such as temperature, viscosity, solubility of gas in liquid, the diffusion rate of dissolved oxygen (DO) in liquid, DO concentration, and vapor pressure, can change the status of cavitation [20,21].

However, sonication for decellularization is not without harmful effects, and inappropriate acoustic power can damage the tissues [19]. Whether sonication is superior to other methods in the decellularization of vascular tissues is currently undetermined. Sonication-treated (2% SDS, 24 h) porcine aorta showed barely formed ECM fibers under a scanning electronic microscope [15]. Sonication added to the decellularization process of larynx tissue but also showed that the tissue became structurally fragile, despite achieving acceptable cellular removal [22]. Thus, further evaluation of the effect of sonication on the matrix of vascular tissues in terms of biomechanical match is important for the development of small-caliber vascular grafts when using sonication-decellularized vascular scaffolds.

In this study, we designed a decellularization protocol for human umbilical arteries (HUAs) utilizing sonication. We evaluated four protocols and compared the results with native umbilical artery histology, DNA amounts, and mechanical properties. The power of sonication was adopted from Azhim’s study [15] and adjusted according to our conventional decellularization protocol [23]. The goal is to reduce the exposure time of detergent and decrease the whole decellularization duration. The criteria for satisfactory decellularization of the graft were defined as no residue visible nuclear components, less than 50 ng DNA/mg dried tissue, and well-preserved ECM structure [24]. In addition, a cell compatibility test of the arterial scaffold was used to assess endothelial cell recellularization. Finally, the optimized arterial scaffold was implanted into a rat abdominal aorta for image and histological evaluation to show that the remaining cellular and intracellular debris in the decellularized human umbilical arterial scaffold would not cause an undesired acute thrombosis or immune response in the recipient.

## 2. Materials and Methods

The study was approved by the Institutional Animal Care and Use Committee of Taipei Veterans General Hospital. All animal care complied with the Guide for the Care and Use of Laboratory Animals. Human tissue was obtained using protocols approved by the Institute Review Board of Taipei Veterans General Hospital. All human subjects signed a consent form approved by the Institute Review Board of Taipei Veterans General Hospital (No. 2013-08-020BC).

### 2.1. Recovery of the Human Umbilical Artery

Human umbilical cords were obtained from the Department of Obstetrics and Gynecology at Taipei Veterans General Hospital, Taipei, Taiwan. The cords were kept at 2–8 °C immediately after delivery, and the overall storage time from harvest to processing did not exceed 24 h. Umbilical vessels approximately 20–30 cm in length were isolated after meticulous removal of Wharton’s jelly under sterile conditions and manipulation. Once one human umbilical vein and two umbilical arteries were separated from the umbilical cord, they were disinfected in an antibiotic cocktail containing 250 μg/mL cefuroxime, 200 μg/mL ciprofloxacin, 80 μg/mL gentamicin, 50 μg/mL vancomycin, 1000 units/mL colistin, and 200 μg/mL amphotericin B in Medium 199 (Invitrogen, Carlsbad, CA, USA) for 24 h at 4 °C. Isolated umbilical arteries were then cut into segments approximately 5 cm in length and flushed with phosphate buffered saline (PBS) a few times.

### 2.2. Decellularization of HUA

The sonication-assisted decellularization process started with agitating the HUA in a sonication tank filled with 2% SDS/PBS buffer. Four sonication power and treatment time conditions were tested: 204 W for 4 h, 204 W for 8 h, 285 W for 2 h, or 285 W for 4 h (n = 6 for each condition). The sonication-assisted decellularization platform is shown in Appendix A. Since ultrasound with electric power leads to higher temperatures that might degrade the ECM, the temperature was strictly monitored and controlled in the range of 30–38 °C. Then, the HUA was shaken in PBS at 80 rpm for 15 min, and the wash buffer was changed twice. The arteries were further immersed in 0.5X PBS containing 12% fetal bovine serum with shaking under 1000 rpm for 48 h at 37 °C, followed by two washes with PBS at 80 rpm for 15 min. The conventional decellularization of HUA was according to our previous protocol [23]. Finally, decellularized HUAs (DHUAs) were preserved at 4 °C for further evaluation. (The conventional decellularized HUAs is abbreviated as cDHUAs. The sonication-assisted decellularized HUAs is abbreviated as sDHUAs).

### 2.3. Histology of HUAs and DHUAs

For histological evaluation of cell removal and scaffold structure, 4 μm-thick transverse sections of HUAs, cDHUAs, and sDHUAs underwent 4,6-diamidino-2-phenylindole (DAPI), hematoxylin and eosin (HE; Sigma-Aldrich, St. Louis, MO, USA), Masson’s trichrome, and elastin Van Gieson’s (EVG, Sigma-Aldrich) staining. Collagen types I and III, laminin, and fibronectin were detected by immunohistochemical staining (IHC) (Collagen type I MAI-26771, 1:1000, Thermo Fisher Scientific (Waltham, MA, USA) and type III PAI-28870, 1:150, Thermo Fisher Scientific and laminin PAI-16730, 1:500, Thermo Fisher Scientific; fibronectin, ab23751, 1:1400, Abcam).

### 2.4. DNA Quantification

DNA quantification was performed using Quant-its Picogreen® dsDNA reagent (Invitrogen, Carlsbad, CA, USA) according to the manufacturer’s instructions. Briefly, lyophilized HUA, cDHUA, and sDHUA were weighed and digested with papain (papaya proteinase I, Sigma) at a concentration of 125 µg/mL at 60 °C overnight. The digested samples were diluted with TE buffer (10 mM Tris-HCl and 1 mM EDTA, pH 7.5; Invitrogen, Waltham, MA, USA) and incubated with PicoGreen reagent. Fluorescence was measured at an excitation wavelength of 485 nm and an emission wavelength of 530 nm using a Spectarmax iD5 Multi-Mode Microplate Reader (Molecular Devices, San Jose, CA, USA). Bacteriophage λ DNA (Invitrogen) was used as a standard [25]. DNA quantification was performed in triplicate.

### 2.5. ECM Component Quantification

The collagen, elastin and glycosaminoglycan (GAG) content of HUA, cDHUA, sDHUA were determined using the Sircol Insoluble Collagen Assay Kit (Biocolor Life Science Assays, Carrickfergus, UK), Fastin Elastin Assay Kit (Biocolor Life Science Assays, Carrickfergus, UK), and the Blyscan Sulfated glycosaminoglycan Assay Kit (Biocolor Life Science Assays, Carrickfergus, UK), according to the manufacturer’s instructions. ECM quantification were performed in quadruplicate.

### 2.6. Mechanical Property Test

The HUAs, cDHUAs, and sDHUAs (each n = 3) were tested for the maximum pressure of the system (200 mmHg). The setting of the mechanical test is shown in Appendix A. The system was filled with Lactated Ringer solution as the flow fluid and pushed by an infusion pump in one direction towards the vessels, which were attached to a flow circuit, submerged in and perfused with medium, and stretched to λ = 1.3 (130% of length prior to attachment). The luminal pressure was cycled between zero and the target pressure, which was incrementally increased by 10 mmHg after three cycles to each target (rate of increase: ~60 mmHg/min; rate of decrease: ~40 mmHg/min). The pressure was dynamically controlled using a sphygmomanometer (Spirit, Taipei, Taiwan). Vessel diameter was observed via light microscopy, with image output by a digital camera. The external diameter was calculated and recorded synchronously with pressure. Pressure-diameter data were used to determine the compliance of the arteries and scaffolds. The mean diameter was calculated along a portion of the vessel length (anterior, middle, and posterior) from the image data. Compliance was defined by the following equation: C = ΔD/ΔP. The test was performed for each scaffold in triplicate.

### 2.7. Cell Seeding Test

Human umbilical vein endothelial cells (HUVECs) were cultured from the umbilical vein as previously described. Passages 2–3 were used in the experiments. Before cell seeding, sDHUAs were immersed in PBS containing 250 μg/mL cefuroxime, 50 μg/mL vancomycin, 1000 U/mL colistin at 4 °C for 24 h. sDHUAs (~1 cm^2^) were cut longitudinally, flattened with the inner face-upward and placed in 6-well plates. HUVECs were seeded on sDHUAs at a density of 1 × 10^6^ cells/cm^2^ in 100 µL endothelial cell growth medium (Cell Applications Inc., San Diego, CA, USA). Cell-seeded scaffolds were then incubated at 37 °C and 5% CO_2_ for 48 h. The HUA, sDHUAs, and decellularized sDHUAs were processed as frozen sections for histological examination (HE, DAPI, and CD31) and SEM to confirm the attachment of the cells. Each experiment was performed in triplicate.

### 2.8. Scanning Electron Microscopy (SEM)

The samples were fixed with 2.5% glutaraldehyde solution (Sigma-Aldrich) at room temperature for 16 h. The samples were washed three times for 10 min each with PBS and then treated with 1% osmium tetroxide (Sigma-Aldrich) for 1 h. After treatment, the samples were rinsed with PBS three times before embedding for SEM (JSM-7600F, JEOL, Ltd.) examination. The specimens were coated with a 10–20 nm-thick platinum layer after critical point drying and examined using SEM.

### 2.9. Biocompatibility Evaluation in a Rat Abdominal Aorta Implantation Model

Before implantation, the freeze-dried sDHUAs were briefly rehydrated by immersion in PBS at room temperature. sDHUAs were then implanted into male SD rats (300–600 g, BioLASCO, Yilan, Taiwan) as an abdominal aorta bridge (n = 4). Briefly, the rats were anaesthetized with an intraperitoneal injection of 50 mg/kg body weight Zoletil 50 (Virbac, Carros, France). The anaesthetized rat was then placed in a supine position over a warm pad. After shaving and sterilization, a midline laparotomy was performed. Then, the abdominal aorta between the infrarenal artery and iliac artery bifurcation was explored after lateralizing the intestine and opening the retroperitoneal fascia. The aorta was carefully dissected from the inferior vena cava, and the side branches were ligated. After clamping proximally and distally, a 1 cm aorta segment was replaced with the sDHUA with end-to-end anastomosis (9–0 nylon, interrupted sutures). No anticoagulation or antiplatelet drugs were administered perioperatively. After the clamps were released, hemostasis was checked, the visceral vessels were returned, and the wound was closed in layers. The rats recovered from anesthesia in a separate cage, where they received food and water ad libitum.

### 2.10. Magnetic Resonance Angiography and Histological Examination

Two rats that underwent sDHUA implantation received MRA to evaluate the patency at day-3 and day-90. Imaging was performed using a 4.7-T magnetic resonance scanner (Biospec 47/40, Bruker BioSpin, Ettlingen, Germany) equipped with an active shielding gradient (20 g/cm in 80 ls). The rat was initially anaesthetized with 5% isoflurane in air at a flow rate of 1 L/min. When fully anaesthetized, the animal was placed in a prone position and fitted with a custom-designed head holder inside the magnet. Anesthesia was then maintained at 1.0–1.2% isoflurane in air at a flow rate of 1 L/min throughout the experiment. Images were acquired using a 72 mm birdcage transmitter coil and a separate quadrature surface coil for signal detection. T2WIs were acquired using a 3D fast SE sequence with a repetition time of 2200 ms, an effective echo time of 33 ms, nex = 4, slice number = 20, slice thickness = 1.5 mm, matrix size = 200 × 200, FOV = 8.5 × 8.5 cm, and scan time = 3 min. After MRA, the rats were sacrificed, and the sDHUAs were explanted for morphological and histological examination. Slices were taken from the mid-portion of the grafts. A 4 μm-thick transverse section of the sDHUAs underwent HE, CD31 (ab182981, 1:100, Abcam, Cambridge, MA, USA), vWF (ab6994, 1:2000, Abcam), eNOs (ab5589, 1:80, Abcam), α-SMA (C6198, 1:5000, Sigma-Aldrich), and CD45 (ab10558, 1:150, Abcam) staining.

### 2.11. Statistical Analysis

Statistical analysis was performed using GraphPad Prism 6.0 software with analysis of variance by two-way ANOVA followed by Bonferroni’s post-test. Results were reported as mean ± standard error. A *p*-value of <0.05 was considered significant.

## 3. Results

### 3.1. Histological Evaluation of HUAs, cDHUAs, and sDHUAS

The results of histology in the native, conventional treated and four sonication-treated groups are shown in Figure 1 and Figure 2. HE staining (Figure 1A–F) showed that the different protocols combined with sonication could remove most cellular components, similarly to conventional decellularization. In all four sonication-assisted decellularization protocols, the intimal layer became less oriented, but the layer structure was preserved. With an increase in sonication power and duration, more cavitation was noted at the subendothelial layer. DAPI staining (Figure 1G–L) confirmed that no cellular signals were observed after decellularization with the different sonication protocols compared with HUA. Masson’s trichrome staining (Figure 1M–R) showed preservation of collagen and fibrous structure in sDHUAs, similarly to cDHUA. EVG staining (Figure 1S–X) revealed that elastin fibers were mostly preserved in the group sonicated at 204 W for 4 h. Less elastin fibers were preserved in the group sonicated at 204 W, 8 h and 285 W, 2 h. Almost no elastin fibers were noted in the group of 285 W, 4 h. IHC staining of sDHUAs showed that types I and III collagen (Figure 2A–L) were preserved as a layer structure at the media layer of sDHUAs treated by sonication at 204 W for 4 h and 8 h. Types I and III collagen were less aligned in the groups treated at 285 W for 2 h and 285 W for 4 h. Fibronectin (Figure 2M–R) was well preserved at the inner surface of sDHUAs treated at 204 W for 4 h and less preserved in the other protocols. Laminin (Figure 2S–X) was mostly preserved in the sDHUAs treated at 204 W for 4 h and 204 W for 8 h, but less preserved in the sDHUAs treated at 285 W for 2 h and 285 W for 4 h. The distribution and alignment of these biomolecules in the group of 204 W for 4 h were most similar to cDHUA. Therefore, treatment of sDHUA from HUA using sonication-assisted decellularization at 204 W for 4 h preserved the intact layer structure, with most components including types I and III collagen, elastin, fibronectin, and laminin, the best among the four protocols.

### 3.2. DNA Quantification of HUAs, cDHUAs, and sDHUAS

In Figure 3a, DNA quantification revealed that the DNA content of HUAs was approximately 392.5 ± 35.3 ng/mg dry weight. The residual DNA amount and percentage of sDHUA relative to HUA were 24.1 ± 8.2 ng/mg dry weight and 6.1%, 179.7 ± 45.9 ng/mg dry weight (45.8%), 48.8 ± 13.8 ng/mg dry weight, (12.4%), and 114.9 ± 2.11 ng/mg dry weight, (29.3%) after sonication with 204 W for 4 h, 204 W for 8 h, 285 W for 2 h, and 285 W for 4 h, respectively. More than 90% of DNA was eliminated from sDHUA by 204 W sonication for 4 h, better than the other sonication-assisted protocols. The residual DNA amount of cDHUAs was 6.2 ± 3.2 ng/mg dry weight (1.5%). There was no significant difference in residual DNA amount between cDHUA and sDHUA.

### 3.3. ECM Component Quantification

In Figure 3b, elastin quantification revealed that the elastin content of HUA, was approximately 304.9 ± 51.45 µg/mg dry weight. The elastin content of each group were 178.85 ± 21.95 µg/mg dry weight (cDHUA), 171.34 ± 55.08 µg/mg dry weight (sDHUA, 204 W, 4 h), 179.88 ± 32.62 µg/mg dry weight (sDHUA, 204 W, 8 h), 184.25 ± 65.84 µg/mg dry (sDHUA, 285W, 2 h), and 142.15 ± 22.19 µg/mg dry weight (sDHUA, 285 W, 4 h), respectively. In Figure 3c, GAG content of HUA, was approximately 19.33 ± 1.66 µg/mg dry weight. The GAG content of cDHUA and each group of sDHUA were 7.27 ± 2.13 µg/mg dry weight, 8.77 ± 1.78 µg/mg dry weight, 8.7 ± 1.61 µg/mg dry weight, 9.32 ± 0.66 µg/mg dry weight, and 8.82 ± 2.02 µg/mg dry weight, respectively. In Figure 3d, collagen content of HUA, was approximately 178.12 ± 21.25 µg/mg dry weight. The collagen content of cDHUA and each group of sDHUA were 291.45 ± 183.57 µg/mg dry weight, 311.61 ± 132.96 µg/mg dry weight, 207.27 ± 75.57 µg/mg dry weight, 274.79 ± 83.65 µg/mg dry weight, and 341.06 ± 277.92 µg/mg dry weight, respectively. There was significant decrease in elastin and GAG content in cDHUA or sDHUAs as compared to HUA. But there was no significant difference in elastin and GAG content between cDHUA and sDHUAs. In addition, most collagen content was preserved in both cDHUA and sDHUAs.

### 3.4. Evaluation of the Mechanical Properties of HUA With Sonication-Assisted Decellularization

During the tests, rupture of DHUAs treated by sonication with 204 W for 8 h, 285 W for 2 h, and 285 W for 4 h was observed. Thus, only DHUAs treated by sonication with 204 W for 4 h completed the triplicate tests. The compliance of sDHUAs (204 W for 4 h) was slightly higher than that of native arteries at pressures of 30, 60, 90, 120, and 150 mmHg. However, no significant difference was noted among HUA, cDHUA, and sDHUA (204 W for 4 h) (Figure 3e). Additional tests in stretch ratio = 1, 1.2, and 1.4 were done, and the results were added to Appendix A.

### 3.5. HUVECs Seeded on Optimized Scaffolds

The DHUAs treated at each condition were seeded with HUVECs, even though the sDHUAs treated with 204 W for 8 h, 285 W for 2 h, and 285 W for 4 h did not withstand the mechanical test, which excluded their suitability for further in vivo implantation tests. After cultivation for 48 h, HE and DAPI staining demonstrated attachment of HUVECs on the scaffolds, as shown in Figure 4 (204 W for 4 h-treated scaffolds shown). Attachment of HUVECs on the DHUAs from 204 W for 8 h, 285 W for 2 h, and 285 W for 4 h treatments were also observed by CD31 and vWF staining (please see the Appendix A). SEM also confirmed the attachment of HUVECs on each group of sDHUAs (Figure 5).

Taking the above together, we found that the sDHUA treated with 204 W for 4 h was the optimal condition in the current study for HUA decellularization, and met all decellularization criteria; therefore, in vivo testing was performed using this sDHUA.

### 3.6. Biocompatibility Evaluation in a Rat Abdominal Aorta Implantation Model

Four rats that received sDHUAs (204 W, 4 h) survived. One was sacrificed at day-3 and three were sacrificed at day-90. The sDHUAs showed slight dilatation in vivo as compared to the graft immediately after anastomosis and release of the clamps (Figure 6A, A: immediately after anastomosis; B: Day 90). MRA showed patency sDHUAs as compared to native rat abdominal aorta (Figure 6C–E: sDHUAs, C: coronal view, D: sagittal view, E: three-dimensional merge; Figure 6F–H: native aorta, F: coronal view, G: sagittal view, H: three-dimensional merge). Histological examination (Figure 7) did not show thrombosis formation at day-3 (Figure 7A–L) and day-90 (Figure 7M–X). However, no obvious endothelial cell (CD31, vWF-stained) attachment at the inner surface of sDHUA was noted at day-3 (Figure 7H,I). There was minimal smooth muscle cell (α-SMA-stained) and inflammatory cell (CD45-stained) infiltration. At day-90, obvious functional endothelial cell (CD31, vWF, eNOs-stained) attachment at the inner surface of sDHUA was noted (Figure 7T–V). Compared to sDHUA at day-3, the CD31+/vWF+ endothelial cells were much more prominent and intact at day-90. In particular, the expression of eNOS in endothelial cells was much more significant at day-90 than that at day-3. Further, under the α-SMA expression, the spindled smooth muscles in the tunica media were more increased at day-90 than those at day-3, while the infiltration of CD45+ inflammatory cells showed no significant difference. The thickness and structure of aortic walls were relatively intact and thicker at day-90, in comparison with those at day-3.

## 4. Discussion

Sonication is a form of energy generated by sound waves at frequencies too high to be detected by the human ear [20]. Sonication has been applied in a variety of fields, according to its power and frequency, to provide detection or sonochemistry [16]. The decellularization effect of sonication could be derived from the effect of ultrasound on the tissues. It is known that acoustic streaming, which is produced by ultrasound, has a bioeffect on cell cytoplasm [26]. In addition, higher power low frequency ultrasound can produce physical or chemical effects in a medium, such as cavitation or emulsion, to disrupt cells. Syazwani et al. demonstrated that sonication using low frequency ultrasound is capable of completely decellularizing the aorta [16]. Azhim et al. also showed that using detergents (2% SDS) as the medium in sonication decellularization could lower the dissolved oxygen (DO) concentration. The lower DO concentration could maximize the cavitation effect, which contributes to significant increases in decellularization efficiency [20].

Sonication is effective in assisting detergent-based decellularization to dissociate the cellular components from ECM but leaves the backbone of collagen/elastin and the connection between collagen- biomolecules intact. It is known that fibronectin or laminin binds to cells through cell surface receptors (integrins) and specifically interacts with other proteins, including collagen, fibrin, and heparin/heparan sulphate [27]. Thus, sonication could be beneficial in selectively dissociating the cells from ECMs and rupture of the cell membrane while preserving the major proteins and biomolecules in the ECMs.

The advantage of using sonication in decellularization is assisted by its various effects on the medium through which it is transmitted. In this study, we found that sonication could achieve cellular membrane rupture and nuclear material release. However, if there is no washing procedure, the nuclear materials will still accumulate in the scaffolds. Another study also showed that DNA appeared as a diffuse smear in HUAs after incubation with 3-[(3-Cholamidopropyl)dimethylammonio]-1-propanesulphonate (CHAPS) and SDS buffers [25]. Residual materials remaining in decellularized tissues may induce a severe inflammatory response upon implantation into a recipient [28]. For adequate nuclear material removal, using serum and a sufficient washing procedure is still necessary. Serum has been shown to be effective in nuclear material removal and was used to remove DNA from the umbilical artery and other tissues after detergent treatment [29]. Thus, we added serial washing steps for the thorough removal of cellular components.

Another benefit of using sonication in decellularization is reducing the SDS treatment time and has been proven to be an effective way of removing native cells. Although SDS is quite effective in removing cell residues from tissue compared to other detergents, it appears to be more disruptive to the ECM [30,31]. Furthermore, SDS does have toxicity to humans or animals, and a formulation containing 5% SDS caused depression, labored breathing, diarrhea, and even to death in four out of 20 animals [32]. It has also been suggested that residual cytotoxic SDS is responsible for the low levels of cell in-growth observed in SDS decellularized tissue. Thus, decrease of detergent concentrations or length of exposure is the strategy to reduce residual cytotoxic SDS in the scaffolds, to avoid adverse effect once in vivo application. However, our cell seeding test did show well HUVEC attachment on the scaffold, which may reflex the adequate washing steps for vascular scaffold to minimize the residual SDS. Gratzer et al. also showed that the level of residual SDS in decellularized tissue could be adequately reduced. They strongly indicated that residual SDS cytotoxicity is not responsible for the low cell re-population in SDS decellularized tissues, but the alterations in tissue structure or biochemical composition [33]. Here, the advantage of adding sonication in SDS decellularization is to afford energy in terms of intensity and frequency, which could help in the balance of SDS concentrations, formation of micelles, penetration of detergent, preservation of the ECM structure, and increase in the efficiency of cellular removal [34,35]. Our results have yielded an improved protocol that uses a shorter duration of SDS treatment (4 h), while being at least as effective in achieving removal of cellular components and DNA of umbilical arterial tissue. The decellularized umbilical arterial scaffold could further support cell-repopulation without obvious cytotoxicity.

The umbilical artery is suitable for the development of small-caliber tissue engineering vascular grafts. However, several challenges remain that need to be addressed. In histomorphology, the umbilical artery has a constricted, branched shape lumen lined by endothelial cells. The lumen is not circular and usually collapses in the transverse direction. In our experience, the problem of irregular lumen could be solved by reshaping the lumen by inserting a rod inside the lumen during the decellularization and freeze-drying process. Another concern is the thick outer layer of umbilical artery. Although thickness of the vascular wall is important for the maintenance of biomechanical properties, the disadvantage is leaving a nonuniform wall thickness while dissecting manually, which may decrease the vessel wall strength. Automated dissection could maintain a uniform wall thickness throughout the vessel length, which, in turn, may maintain morphological characteristics, and does not affect its biomechanical behavior [36,37].

As for the layer structure of the vessel, the intimal layer of the umbilical artery is usually thickened, and there are thin, elongated, or wavy smooth muscle cells (SMCs) residing in the subendothelial layer. The internal elastic lamina at the intima-medial junction is not prominent [38,39]. The media was usually thick, double the size of adventitia and constituted by SMCs, elastin, and collagen fibers. The inner SMCs were longitudinally aligned, while the outer SMCs were crossing spiraled. The intermuscular spaces were occupied by mucopolysaccharides [39]. Types I and III collagen were found to be the most abundant collagens in the umbilical artery. The tunica adventitia consists of collagen, elastin, and SMCs [40]. In our study, we observed a similar histological pattern in the native umbilical artery with expression of types I and III collagen in the media and fibronectin and laminin in the subendothelial layer.

After sonication treatment, the layer structure at the media layer could still be maintained. However, tissue treated with high power or long duration showed an increase in cavity formation and layer dissociation in the vessel wall (as shown by the group 204 W 8 h, 285 W 2 h, and 285 W 4 h in Figure 1 and Figure 2). It is known that sonication can evoke a cavitation effect, which generates localized mechanical and chemical energy [41]. As more cavitation bubbles are formed during sonication, many molecules are dissociated [20]. Our findings showed that high power or longer duration of sonication could disrupt the main structural fibers of the umbilical artery histologically. In addition, higher sonication power could lead to adverse effects on the vascular tissues, for example, free radical reactions, shock waves, shear stress, and microjet [42]. However, small biomolecules, such as fibronectin and laminin, were not severely eliminated by sonication and could be preserved in the scaffolds and support attachment of ECs.

The normal pressure at which the umbilical arteries are exposed in vivo is approximately 50 mmHg [43]. However, higher pressure exposure is expected for use as a vascular conduit. The compliance of DHUAs (204 W, 4 h) did not show a significant difference compared with the native umbilical artery from 30 to 120 mmHg. In addition, we tested the compliance at a 1.3 stretch ratio for the simulation of in vivo conditions [44]. However, the general compliance of DHUAs is still higher than that of native umbilical artery. In the literature, the results of the mechanical properties of decellularized HAUs were mixed [25,45]. Although high-power sonication could assist cellular removal as in the groups of 204 W for 8 h, 285 W for 2 h, and 285 W for 4 h, these groups of sDHUAs already had major structural insufficiencies and could not withstand the perfusion mechanical tests. Despite the different decellularization protocols, the mixed results in the mechanical properties of decellularized HUAs could be attributed to the contribution of Wharton’s jelly, since it is largely responsible for the strength of the blood vessel [46]. It could also be related to the thickness of the arterial wall after manual removal of Wharton’s jelly. In addition, overly aggressive dissection could result in compliance increasing as the vessel wall becomes thin and weak [37].

The rat aorta implantation was performed to demonstrate the biocompatibility of sDHUA, as shown by providing patency of blood flow and limited inflammation in vivo. Endothelialization usually takes weeks to complete in decellularized vascular grafts, and the mechanism is complex [47]. Even in decellularized allograft transplantation, little repopulation was noted two weeks after implantation, and reendothelialization was noted only in the peri-anastomotic regions eight weeks after implantation [48]. Thus, it is expected to take a longer time for obvious reendothelialization in decellularized xenogenic scaffolds in vivo in consideration of species discrepancy [49]. For cell repopulation, transmural migration could be a major pathway in our observation, as shown by smooth muscle cell infiltration in the media of sDHUA. Another pathway could be the recruitment of circulating monocytes from peripheral blood flow (fall-out). Different circulating cells involved in the host inflammatory response may contribute to endothelialization [50]. The role of mononuclear cells in the degradation of ECM scaffolds is also important [51,52]. Furthermore, matrix metalloproteinases (MMPs), such as MMP-2 and MMP-9, are crucial in the degradation of ECM in xenogenic models and may contribute to aneurysmal dilatation [53]. However, monocyte-derived macrophages differentiated on decellularized matrices could release more MMP-2 and not necessarily be stimulated into the activated, inflammatory phenotype [54]. At present, we only observed minimal CD45+ lymphocyte infiltration in the sDHUA to exclude severe rejection, but whether the current phenomenon is beneficial or harmful for sDHUA in regeneration and remodeling may need further longer-term evaluation.

Briefly, our study found that using sonication assisted tissue decellularization could be effective but should be performed with caution. Sonication could have several effects on the tissues. High energy and long duration of treatment could result in tissue fragility, leakage, and weakness. Through our systemic evaluation, we demonstrated that the HUA could be decellularized efficiently by sonication at 40 kHz and 204 W for 4 h followed by washing steps. The DNA component could be eliminated by more than 90%, while the major proteins were preserved. The sDHUAs could be recellularized with HUVECs in vitro and quickly recruited host inflammatory cells after short-term implantation. The results were correlated to both the preservation of biochemical and mechanical properties in HUAs, while eliciting minimal immune reactions. Thus, the optimized sonication-treated HUA showed promise as an ideal scaffold for small-caliber vascular grafts.

## Figures and Tables

**Figure 1 polymers-13-01699-f001:**
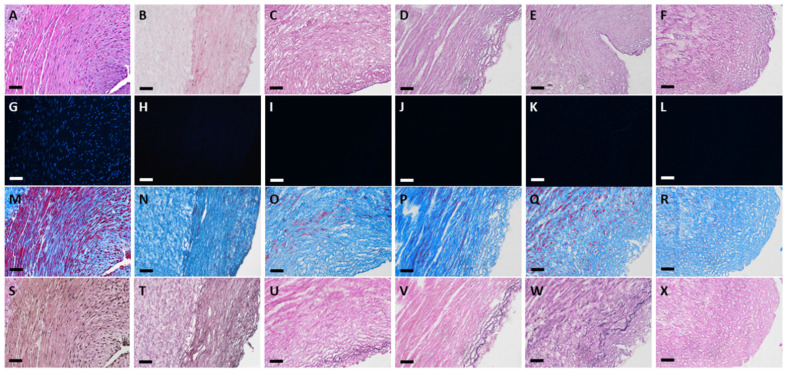
Histological stain of HUA, cDHUA, and sDHUA. (**A**,**G**,**M**,**S**) HUA; (**B**,**H**,**N**,**T**) cDHUA; (**C**,**I**,**O**,**U**) sDHUA, 204 W for 4 h; (**D**,**J**,**P**,**V**) sDHUA, 204 W for 8 h; (**E**,**K**,**Q**,**W**) sDHUA, 285 W for 2 h; (**F**,**L**,**R**,**X**) sDHUA, 285 W for 4 h; (**A**–**F**) H&E stain; (**G**–**L**) DAPI; (**M**–**R**) Masson’s trichrome stain; (**S**–**X**) Elastin Van Gieson’s stain; Magnification: 200×, Scale bar = 50 μm.

**Figure 2 polymers-13-01699-f002:**
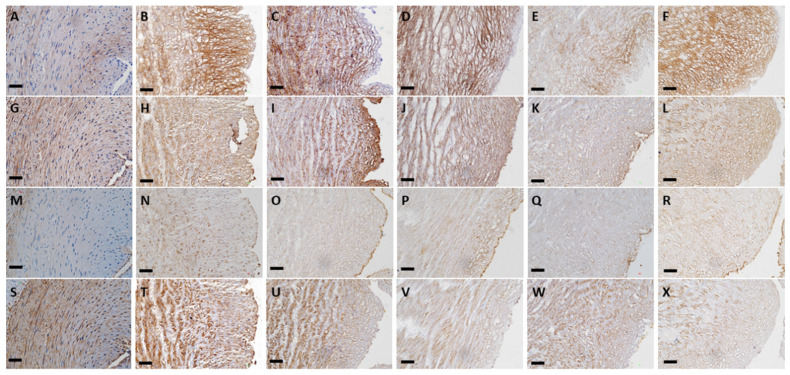
IHC stain of HUA, cDHUA, and sDHUA. (**A**,**G**,**M**,**S**) HUA; (**B**,**H**,**N**,**T**) cDHUA; (**C**,**I**,**O**,**U**) sDHUA, 204 W for 4 h; (**D**,**J**,**P**,**V**) sDHUA, 204 W for 8 h; (**E**,**K**,**Q**,**W**) sDHUA, 285 W for 2 h; (**F**,**L**,**R**,**X**) sDHUA, 285 W for 4 h; (**A**–**F**) Collagen type I; (**G**–**L**) Collagen type III; (**M**–**R**) Fibronectin; (**S**–**X**) Laminin; Magnification: 200×, Scale bar = 50 μm.

**Figure 3 polymers-13-01699-f003:**
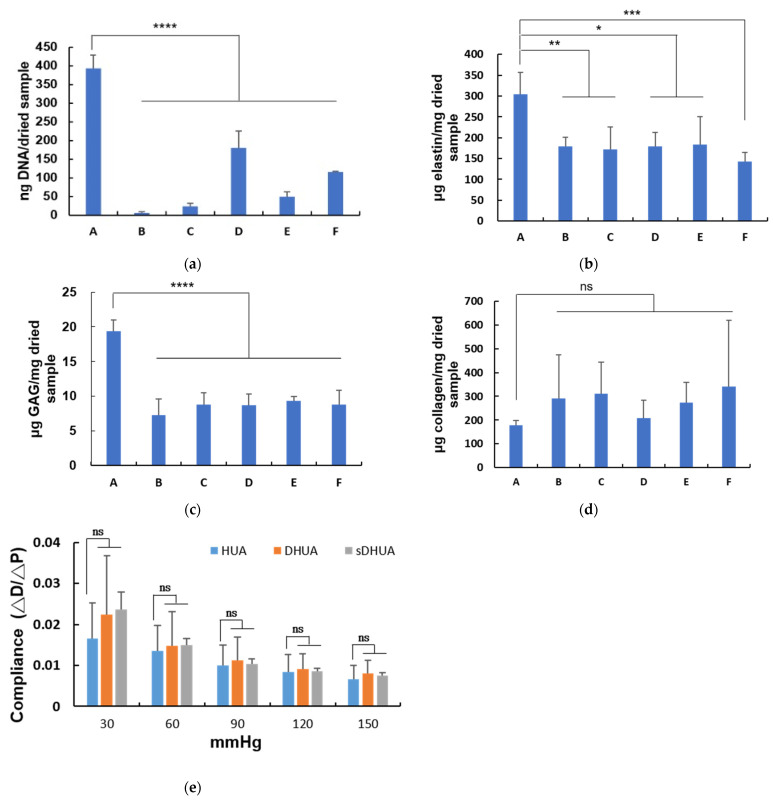
(**a**–**e**): DNA, elastin, GAGs, and collagen content of sDHUA compared with HUA and cDHUA. (A:HUA, B:cDHUA, C:sDHUA, 204 W for 4 h, D:sDHUA, 204 W for 8 h, E:sDHUA, 285 W for 2 h, F:sDHUA, 285 W for 4 h). (**a**) DNA content (n = 3); (**b**) elastin content (n = 4); (**c**) GAG content (n = 4); (**d**) collagen content (n = 4). (**e**) Mechanical property (compliance) between HUA, cDHUA, and sDHUA at different pressure. (λ = 1.3). *: *p* < 0.05, **: *p* < 0.01, ***: *p* < 0.005, ****: *p* < 0.001, ns: no significance.

**Figure 4 polymers-13-01699-f004:**
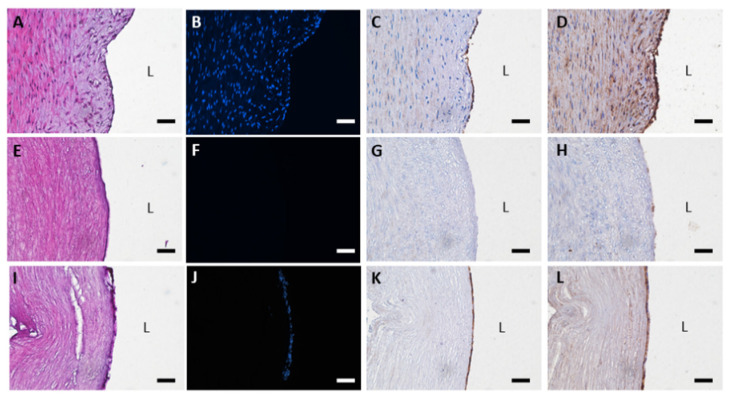
Histological stains of cell seeding sDHUA, 204W for 4 h. (**A**–**D**) HUA; (**E**–**H**) sDHUA; (**I**–**L**) sDHUA seeded with HUVEC; (**A**,**E**,**I**) H&E stain; (**B**,**F**,**J**) DAPI; (**C**,**G**,**K**) CD31; (**D**,**H**,**L**) vWF; Magnification: 200×, Scale bar = 50 μm. L: lumen.

**Figure 5 polymers-13-01699-f005:**
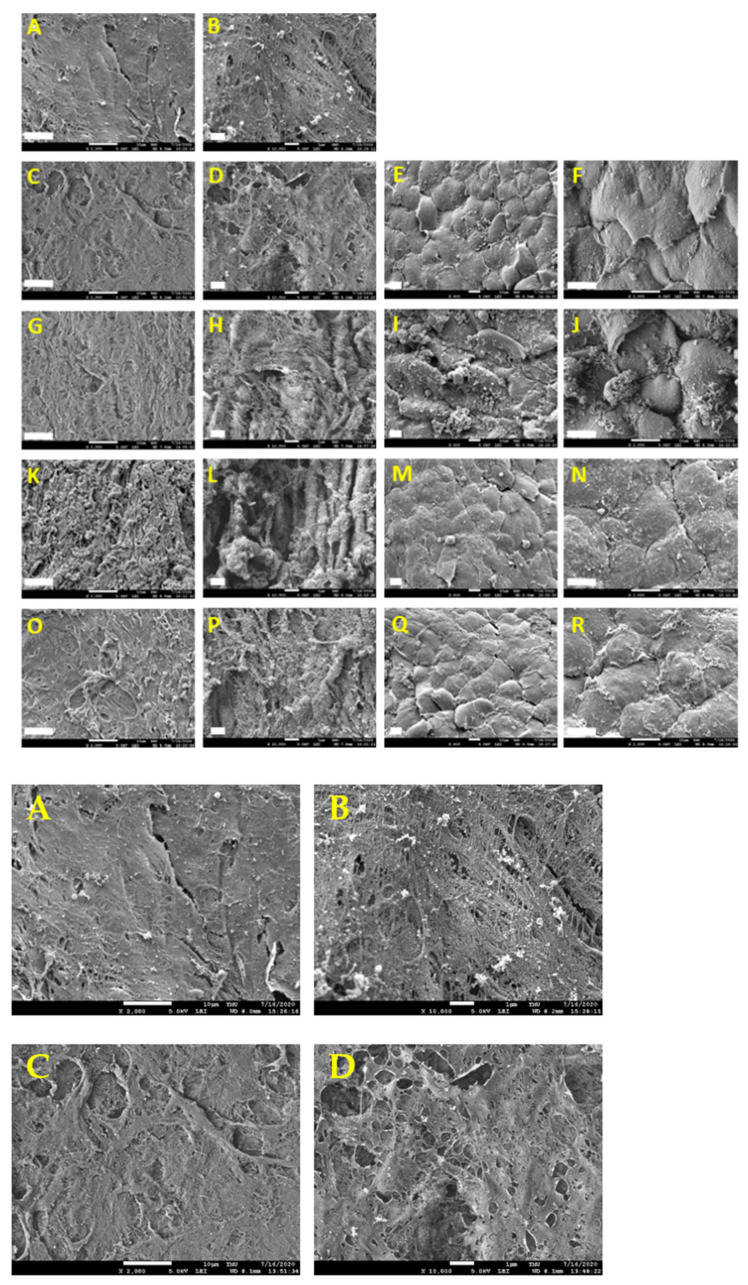
SEM of the luminal surface of HUA and sDHUAs seeded with HUVECs. (**A**,**B**) HUA, (**C**–**F**) sDHUA, 204 W, 4 h, (**G**–**J**) sDHUA, 204 W, 8 h, (**K**–**N**) sDHUA, 285 W, 2 h, (**O**–**R**) sDHUA, 285 W, 4 h. (**A**,**C**,**G**,**K**,**O**) Magnification: 2000×, Scale bar = 10 μm; (**B**,**D**,**H**,**L**,**P**) Magnification: 10,000×, Scale bar = 1 μm; (**E**,**I**,**M**,**Q**) Magnification: 800×, Scale bar = 10 μm; (**F**,**J**,**N**,**R**) Magnification: 2000×, Scale bar = 10 μm.

**Figure 6 polymers-13-01699-f006:**
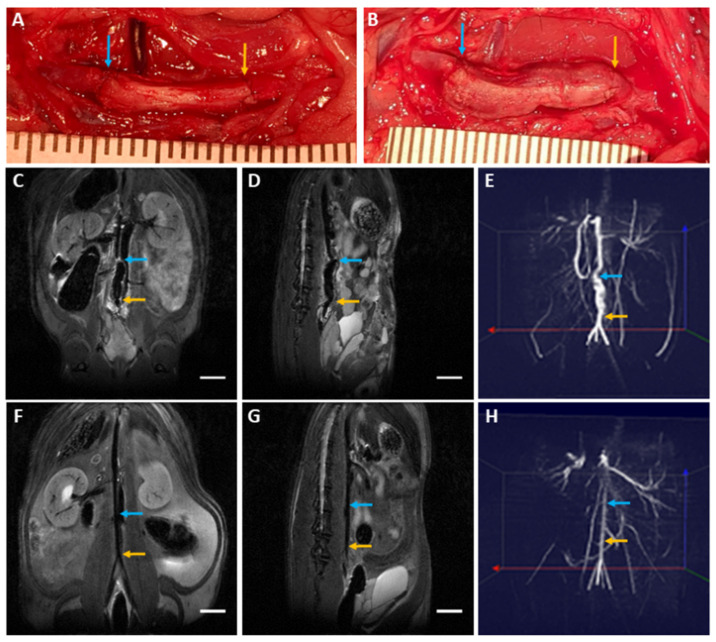
sDHUA implantation and MRI image. (**A**) Gross appearance of the implantation grafts at day-0. (**B**) Gross appearance of the implantation grafts at day-90. (**C**) T2 image of coronal view of implantation graft in rat abdominal cavity. (**D**) T2 image of sagittal view of implantation graft in rat abdominal cavity. (**E**) TOF 3D display of abdominal artery with the implantation graft. (**F**) T2 image of coronal view of naïve rat abdominal cavity. (**G**) T2 image of sagittal view of naïve rat abdominal cavity. (**H**) TOF 3D display of naïve abdominal artery. (**C**,**D**,**F**,**G**) Scale bar = 1 cm; Arrow: anastomosis, blue: near the head, yellow: near the tail.

**Figure 7 polymers-13-01699-f007:**
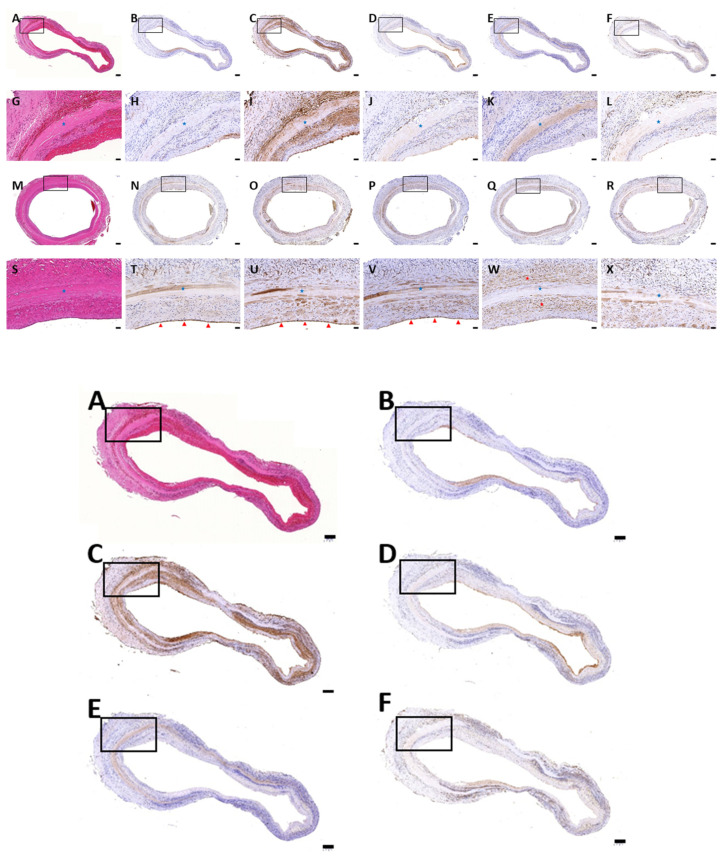
H&E and IHC staining of the explanted grafts at day-30 and day-90. (**A**–**L**) day-3; (**M**–**X**) day-90; (**A**,**G**,**M**,**S**) H&E; (**B**,**H**,**N**,**T**) CD31; (**C**,**I**,**O**,**U**) vWF; (**D**,**J**,**P**,**V**) eNOS; (**E**,**K**,**Q**,**W**) α-SMA; (**F**,**L**,**R**,**X**) CD45; Red arrow heads in (**T**,**U**,**V**) indicated endothelial cells at inner surface, while red stars marked in W indicated α-SMA-stained cells around sDHUA at day-90. The blue stars indicated sDHUA site. Magnification: (**A**–**F**), (**M**–**R**) 40×, Scale bar = 200 μm; (**G**–**L**,**S**–**X**) 200×, Scale bar = 50 μm.

## Data Availability

Not applicable.

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
