# Peer review of "Sonication-Assisted Method for Decellularization of Human Umbilical Artery for Small-Caliber Vascular Tissue Engineering"

_polymers, 2021, doi:10.3390/polym13111699_

Round 1

Reviewer 1 Report

A subsection about the real applications need to be included.

Is there any impact of SDS on the health?

Figures need to be prepared in a better was.

Figures 5 need better resolution

Author Response

Dear reviewer:

Thank you very much for your time and providing valuable suggestions to improve the quality of our manuscript. The response to your recommendation were listed as below and the context was revised point-by-point and highlighted in red, according to your suggestion.

Q1: A subsection about the real applications need to be included.

Ans: We added a subsection about the real application of small-caliber vascular graft at the first section of introduction, line 49-61. The added context and reference (ref. 1-7) were highlighted in red.

Q2: Is there any impact of SDS on the health?

Ans: Overdosage of residual SDS could have impact on the humans or animals. But after shortening SDS treatment duration and lowering concentration under sonication assistant, the HUVECs could adhere on the decellularized umbilical arterial scaffold, which meant very low cytotoxicity. We may consider evaluate residual SDS in the scaffold and thanks again for your recommendation. A section of discussion about the impact of SDS on the health is added in the discussion and highlighted in red.(line 619-630, added reference 33,34)

Q3: Figures need to be prepared in a better was.

Ans: Thanks for the suggestion. We’ve adjusted the figure 3 and provided better resolution of figure 5.

Q4: Figures 5 need better resolution

Ans: We’ve provided better resolution of figure 5 at line 366-461, please check it.

Reviewer 2 Report

Journal: Polymers (ISSN 2073-4360)

Manuscript ID:polymers-1203362

Type: Article

Number of Pages:29

Title: Sonication-assisted method for decellularisation of human umbilical artery for small-calibre vascular tissue engineering

Authors: Chih-Hsun Lin, Kai Hsia,Chih-Kuan Su,Chien-Chin Chen,Chang-Ching Yeh,Hsu Ma, Jen-Her Lu

Revision:

The Article entitled: "Sonication-assisted method for decellularisation of human umbilical artery for small-calibre vascular tissue engineering" is very interesting. The authors found that the optimised sonication treated human umbilical artery (HUA) showed promise as an ideal scaffold for small-calibre vascular grafts. They demonstrated that the HUA could be decellularised efficiently by sonication at 40 kHz and 204 W for 4 h followed by  washing steps. The sDHUAs could be recellularised with HUVECs in  vitro and quickly recruited host inflammatory cells after short-term implantation.

I think that the manuscript is well written and suitable for its publication in Polymers journal, after minor revision:

Minor revision:

There are two figure 3 (lines: 315-320 and 321-323) and two fgure 4 (lines 337-339 and 340-346): check the names of the figures. In particular the first figure 3 (lines 315-320), the graphs appear incomplete. They were probably loaded in a format not suitable for full viewing.

Author Response

Dear reviewer

Thank you very much for your time and providing valuable suggestions to improve the quality of our manuscript. The response to your recommendation were listed as below and the context was revised point-by-point and highlighted in red, according to your suggestion.

Question: There are two figure 3 (lines: 315-320 and 321-323) and two figure 4 (lines 337-339 and 340-346): check the names of the figures. In particular the first figure 3 (lines 315-320), the graphs appear incomplete. They were probably loaded in a format not suitable for full viewing.

Ans: After revision, line 333-334 showed figure 3, line 335-337 showed supplement figure 3. The graphs in figure 3 were adjusted to appear completely. Line 352-353 showed figure 4 and line 359-360 showed supplement figure 4.